# Using Standardized Time Series Land Cover Maps to Monitor the SDG Indicator "Mountain Green Cover Index" and Assess Its Sensitivity to Vegetation Dynamics

**Lorenzo De Simone** [1,*] **, Dorian Navarro** [1]**, Pietro Gennari** [1]**, Anssi Pekkarinen** [2] **and Javier de Lamo** [2]

1 Office of the Chief Statistician, Food and Agriculture Organization of the United Nations, 00153 Rome, Italy; doriankalamvrezos.navarro@fao.org (D.N.); Pietro.gennari@fao.org (P.G.)

2 Forestry Division, Food and Agriculture Organization of the United Nations, 00153 Rome, Italy; Anssi.pekkarinen@fao.org (A.P.); javier.delamorodriguez@fao.org (J.d.L.)

* Correspondence: lorenzo.desimone@fao.org

**Abstract:** SDG indicators are instrumental for the monitoring of countries' progress towards sustainability goals as set out by the UN Agenda 2030. Earth observation data can facilitate such monitoring and reporting processes, thanks to their intrinsic characteristics of spatial extensive coverage, high spatial, spectral, and temporal resolution, and low costs. EO data can hence be used to regularly assess specific SDG indicators over very large areas, and to extract statistics at any given subnational level. The Food and Agriculture Organization of the United Nations (FAO) is the custodian agency for 21 out of the 231 SDG indicators. To fulfill this responsibility, it has invested in EO data from the outset, among others, by developing a new SDG indicator directly monitored with EO data: SDG indicator 15.4.2, the Mountain Green Cover Index (MGCI), for which the FAO produced initial baseline estimates in 2017. The MGCI is a very important indicator, allowing the monitoring of the health of mountain ecosystems. The initial FAO methodology involved visual interpretation of land cover types at sample locations defined by a global regular grid that was superimposed on satellite images. While this solution allowed the FAO to establish a first global MGCI baseline and produce MGCI estimates for the large majority of countries, several reporting countries raised concerns regarding: (i) the objectivity of the method; (ii) the difficulty in validating FAO estimates; (iii) the limited involvement of countries in estimating the MGCI; and (iv) the indicator's limited capacity to account for forest encroachment due to agricultural expansion as well as the undesired expansion of green vegetation in mountain areas, resulting from the effect of global warming. To address such concerns, in 2020, the FAO introduced a new data collection approach that directly measures the indicator through a quantitative analysis of standardized land cover maps (European Space Agency Climate Change Initiative Land Cover maps—ESA CCI-LC). In so doing, this new approach addresses the first three of the four issues, while it also provides stronger grounds to develop a solution for the fourth issue—a solution that the FAO plans to present to the Interagency and Expert Group on SDG Indicators (IAEG-SDG) at its autumn 2021 session. This study (i) describes the new approach to estimate the MGCI indicator using ESA's CCI-LC and products, (ii) assesses the accuracy of the new approach; (iii) reviews the limitations of the current SDG indicator definition to monitor progress towards SDG 15.4; and (iv) reflects on possible further adjustments of the indicator methodology in order to address them.

**Keywords:** SDG 15.4.1; MGCI; land cover; ESA CCI; GLC-LC100

## 1. Introduction

The Food and Agriculture Organization of the United Nations (FAO) is the custodian agency for 21 SDG indicators and, as such, is responsible for collecting data from countries, checking quality and consistency, and reporting national, regional, and global figures to the global SDG database. In addition, the FAO provides technical assistance supporting

countries' efforts in computing and reporting the SDG indicators. In this regard, the FAO has consistently supported the adoption of earth observation (EO) data and geospatial information as an important instrument to effectively measure a range of SDG indicators. Though not a panacea for solving all the challenges related to the immense SDG data needs, as often heralded, EO data can significantly contribute both directly and indirectly to improving the availability, quality, and consistency of SDG indicators. This has been clearly recognized by the UN General Assembly [1] and is also reflected in the establishment of several EO coordination bodies such as the Group on Earth Observations (GEO), the United Nations Committee of Experts on Global Geospatial Information Management (UN-GGIM), the IAEG-SDG Working Group on Geospatial Information, and the Expert Group on the Integration of Statistical and Geospatial Information.

Among the list of SDG indicators for which geospatial information can be used to provide direct measurement is SDG indicator 15.4.2, also known as the Mountain Green Cover Index (MGCI). The MGCI monitors land cover changes in mountain areas to determine which proportion of this area is covered by green vegetation (forest, shrubs and pastureland, and cropland), based on the notion that vegetation is positively correlated with the state of the health of mountains, and, therefore, with their capacity to fulfill critical ecosystem roles [2]. An exception to this rule is green vegetation emerging from areas previously occupied by perennial ice and snow, as a result of climate change-related global warming effects.

In 2016, the FAO developed the first version of the methodology to calculate the MGCI, which involved the juxtaposition of land cover data interpreted with 120,000 plots from satellite data collected through a systematic sampling approach using the FAO Open Foris Collect Earth tool, and the global map of mountain elevation ranges as defined by Kapos et al. in 2000 [3]. The area of green vegetation over total mountain area was calculated based on the proportion of each land cover class in the Collect Earth plot survey over the total mountain area.

Based on this methodology, in 2017, the FAO carried out a global baseline MGCI assessment, producing estimates at the global, regional, and national level. Since these estimates were generated by the FAO itself rather than by countries, the FAO submitted them for validation to National Statistical Offices (NSOs) in compliance with the IAEG-SDG Guidelines on Data Flows and Global Reporting [4]. During this validation process, some countries expressed concerns about: (i) the objectivity of the method; (ii) the data source for land cover types, particularly what was perceived by certain countries as the FAO "imposing a data source"; (iii) the interpretation of the indicator, particularly the assumption that increasing green mountain cover is always a positive change; (iv) the validation process itself, which was cumbersome for both countries and the FAO, particularly due to the very large number of data points to be interpreted and validated by countries.

To address such concerns, in 2020, the FAO reviewed the default calculation approach with the final aim of providing countries with a new EO-based solution for reporting on this SDG indicator, which would be objective and accurate, standardized, simple to implement and validate, sustainable in the long term, and one that allows countries to use both publicly available EO data as well as their own national datasets.

An essentially land cover-based indicator, the MGCI can be assessed using remote sensing imagery and land cover maps via automated algorithms that classify land cover types, rather than by using visual interpretation. The availability of free and open EO data and online cloud computing platforms further facilitates the automation of the reporting process. Similar approaches have recently been adopted by both researchers and National Statistical Offices. Bian et al. [5] used Landsat observations to estimate the green vegetation cover in mountain areas by analyzing the Normalized Difference Vegetation Index (NDVI) using a frequency- and phenology-based algorithm (the algorithm and the data are expected to be made publicly available soon). The National Statistics Offices of Germany and Mexico [6,7], among others, make operational use of high-resolution national

land cover maps at a 10 m spatial resolution to measure and report the official MGCI statistics. However, not all countries have their own national land cover map time series, and different land cover classification systems may exist at the national level, which may require further adjustment of the data to comply with the standard classification used by the MGCI, i.e., the land use/land cover classes of the Intergovernmental Panel on Climate Change (IPCC).

The availability of global yearly land cover maps has significantly increased in recent years. In 2017, the European Space Agency (ESA) made available an archive of consistent global land cover maps at a 300 m spatial resolution on an annual basis from 1992 to 2018 under the framework of the Climate Change Initiative Land Cover (CCI-LC) project [8], In 2019, the Copernicus Climate Change Service (C3S) produced three new global land cover maps, consistent with the ESA maps, on an annual basis from 2016 to 2018. More recently, the Copernicus Global Land Monitoring Service (CGLS) released the Moderate Dynamic Land Cover (GLC-LC100) product [9], a new set of annual global land cover time series from 2015 to 2019 of higher spatial resolution and accuracy than the ESA CCI-CL product. This product is expected to be updated on a yearly basis. Such abundancy of consistent global land cover maps provides an unprecedented opportunity to improve the consistency and expand the temporal coverage of the MGCI, based on a standardized approach that relies on freely available products, which can therefore guarantee a sustainable monitoring effort by countries.

The purpose of this paper is to assess the applicability that these new cutting-edge EO products offer to compute SDG indicator 15.4.2—the MGCI—as well as assessing their limitations. This study therefore (i) describes the new default calculation approach to estimate the MGCI indicator using the ESA's CCI-LC and GLC-LC100 products, (ii) assesses the accuracy of the new approach; (iii) describes the MGCI results at a global, regional, and national level obtained with the new calculation approach; and (iv) reviews the remaining limitations of the current SDG indicator definition to monitor progress towards SDG 15.4, and reflects on possible further adjustments of the indicator methodology in order to address them.

## 2. Materials and Methods

### 2.1. Definition of the MGCI

The official definition of the Mountain Green Cover Index is provided in the metadata of SDG indicator 15.4.2 maintained by the FAO and available either in the global metadata repository or the FAO SDG indicator portal [2]. The MGCI is defined as the ratio of the mountain green cover area to the total mountain area:

$$MGCI = \frac{Mountain\ Green\ Cover\ Area}{Total\ Mountain\ Area} \qquad (1)$$

where: Mountain Green Cover Area = sum of mountain area (Km$^2$) covered by cropland, grassland, forestland, shrubland, and wetland, as defined based on the IPCC classification; *Total Mountain Area* = total area (Km$^2$) of mountains. In both the numerator and denominator, *Mountain* is defined according to Kapos et al. in 2000 [3].

In the next sub-section, we present the principal data sources for the numerator and denominator of the MGCI formula, starting with the denominator.

### 2.2. The MGCI Denominator: Mountain Classification Data—Elevation Layer

The global mountain classification data were obtained from the FAO Mountain Partnership website [10]. The data are based on the UNEP-WCMC mountain classification system [3]; it is a raster dataset, with a 500 m spatial resolution and is derived from the Global 30 Arc-Second Elevation (GTOPO 30). The altitude, slope, and local elevation range (LER) are the criteria used for the mountain classification (Table 1).

**Table 1.** UNEP-WCMC criteria for mountain classes.

| Class | Elevation | Slope | LER 7 km Radius |
|---|---|---|---|
| 1 | >4500 m | Not used | Not used |
| 2 | 3500–4500 m | Not used | Not used |
| 3 | 2500–3499 m | Not used | Not used |
| 4 | 1500–2499 m | >2° | Not used |
| 5 | 1000–1499 m | >5° | OR > 300 m |
| 6 | 300–999 m | Not used | >300 m |

*2.3. The MGCI Numerator: Mountain Green Cover Area*

The baselines for mountain green cover data at national, regional, and global levels were initially estimated by the FAO for the year 2017 using the FAO Open Foris Collect Earth tool powered by Google Earth [11,12]. This approach relied on the visual interpretation of a predefined global sample of 120,000 plots from satellite data.

Instead, the newly proposed spatially explicit approach focuses on the extraction of mountain green cover area directly from standardized land cover maps with a view to automating the process as new land cover maps are made available on a yearly basis by official sources. The FAO identified two key sources for this purpose: the European Space Agency's Climate Change Initiative Land Cover maps (ESA CCI-LC) and the Copernicus Global Land Monitoring Service's (CGLS) Moderate Dynamic Land Cover (GLC-LC100) map. Such new methodology introduces two key benefits: (1) possibility for automation and (2) use of readily available national products for deriving the indicator values.

The European Space Agency's CCI-LC dataset consists of global land cover maps available on an annual basis from 1992 to 2018. The CCI-LC legend contains 37 land cover classes as shown in columns *a* and *b* in Table 2. Such classes have been defined using the Land Cover Classification System (LCCS) developed by the FAO [13]. The spatial resolution of the ESA CCI-LC product is 0.002778° which corresponds to approximately 300 m at the equator. The fact that these maps have a good temporal coverage is the main reason that the FAO selected them as the default data source for mountain green cover area, despite their lower spatial resolution.

**Table 2.** ESA CCI-LC land cover codes (a) and (b) description; corresponding IPPC class (c); corresponding green or non-green label (d).

| (a) Class Code | (b) Description | (c) Reclassification to IPPC | (d) Reclassification to Green/Non-Green |
|---|---|---|---|
| 50 | Tree cover shrub herbaceous cover (>50%)/cropland (<50%) | | |
| 60 | Tree cover broadleaved evergreen closed to open (>15%) | | |
| 61 | Tree cover broadleaved deciduous closed to open (>15%) | | |
| 62 | Tree cover broadleaved deciduous closed (>40%) | | |
| 70 | Tree cover broadleaved deciduous open (15–40%) | | |
| 71 | Tree cover needle leaved evergreen closed to open (>15%) | Forest | Green |
| 72 | Tree cover needle leaved evergreen closed (>40%) | | |
| 80 | Tree cover needle leaved evergreen open (15–40%) | | |
| 81 | Tree cover needle leaved deciduous closed to open (>15%) | | |
| 82 | Tree cover needle leaved deciduous closed (>40%) | | |
| 90 | Tree cover needle leaved deciduous open (15–40%) | | |
| 100 | Mosaic tree and shrub (>50%)/herbaceous cover (<50%) mixed leaf type (broadleaved and needle leaved) | | |
| 10 | Cropland | | |
| 11 | Herbaceous cover rainfed | | |
| 12 | Tree or shrub cover | | |
| 20 | Cropland | | |
| 30 | Mosaic cropland (>50%)/natural vegetation (tree irrigated or post-flooding) | Cropland | Green |
| | Mosaic herbaceous cover (>50%)/tree and shrub (<50%) | | |
| 110 | Mosaic natural vegetation, tree, shrub, herbaceous cover (>50%)/cropland | | |

**Table 2.** *Cont.*

| (a) Class Code | (b) Description | (c) Reclassification to IPPC | (d) Reclassification to Green/Non-Green |
|---|---|---|---|
| 40 | Mosaic natural vegetation (tree shrub herbaceous cover) (<50%) | | |
| 120 | Shrubland | | |
| 121 | Shrubland evergreen | | |
| 122 | Shrubland deciduous | Grassland | Green |
| 130 | Grassland | | |
| 140 | Lichens and mosses | | |
| 160 | Tree cover flooded fresh/saline/brackish water | | |
| 170 | Tree cover flooded fresh or brackish water | Wetland | Green |
| 180 | Shrub or herbaceous cover flooded saline water | | |
| 150 | Sparse vegetation (tree | | |
| 151 | Sparse tree (<15%) shrub herbaceous cover) (<15%) | | |
| 152 | Sparse shrub (<15%) | | |
| 153 | Sparse herbaceous cover (<15%) | | |
| 200 | Bare areas | Other land | Non- |
| 201 | Consolidated bare areas | | green |
| 202 | Unconsolidated bare areas | | |
| 210 | Water bodies | | |
| 220 | Permanent snow and ice | | |
| 190 | Urban areas | Settlement | Non-green |

In contrast to the ESA CCI-LC, the Copernicus Global Land Service (CGLS) delivers annual dynamic global land cover maps at a 100 m spatial resolution (CGLS-LC100) starting from reference year 2015. The CGLS land cover product is characterized by a three-level land cover classification scheme. A detailed legend is shown in Table A1 in Appendix A. In addition to the discrete land cover classes, CGLS-LC100 also delivers continuous field layers named "fraction maps" for each basic land cover class. Such fractional maps provide proportional estimates for vegetation/ground cover for specific land cover types. Discrete CGLS-LC100 maps have been used in this study for the sole purpose of comparing mountain green cover area estimates to date with those derived from ESA CCI-LC, due to the unavailability of CGLS-LC100 within the desired historical timespan for SDG reporting purposes (2000 to present).

*2.4. The New MGCI Approach*

The newly proposed calculation approach focuses on the extraction of mountain green cover data directly from standardized land cover maps with a view to automating the process and avoid the use of visual interpretation.

Specifically, the first step in calculating the overall MGCI value involved the resampling of the elevation range layer at a 500 m resolution to the 300 m resolution of the CCI-LC layer. Secondly, the land cover maps for the years 2000, 2010, 2015, and 2018 were reclassified by mapping the 37 land cover classes featured in the CCI-LC to the six corresponding IPCC land use/land cover classes, four of which are categorized as green (forestland, cropland, grassland, wetland) and two of which are considered non-green (other land, settlement). Results are shown in columns *c* and *d* in Table 2.

The resulting green/non-green binary maps were subjected to a zonal statistics function, and the count of pixels for each class within each UNEP-WCMC mountain class was calculated for each country and territory, defined by the Global Administrative Unit Layers (GAUL). The MGCI Formula (1) was applied at the elevation range level, at national, regional, and global level for the years 2000, 2010, 2015, and 2018. Additionally, disaggregated land cover and elevation range data were calculated following the same approach. The full workflow is shown in Figure 1.

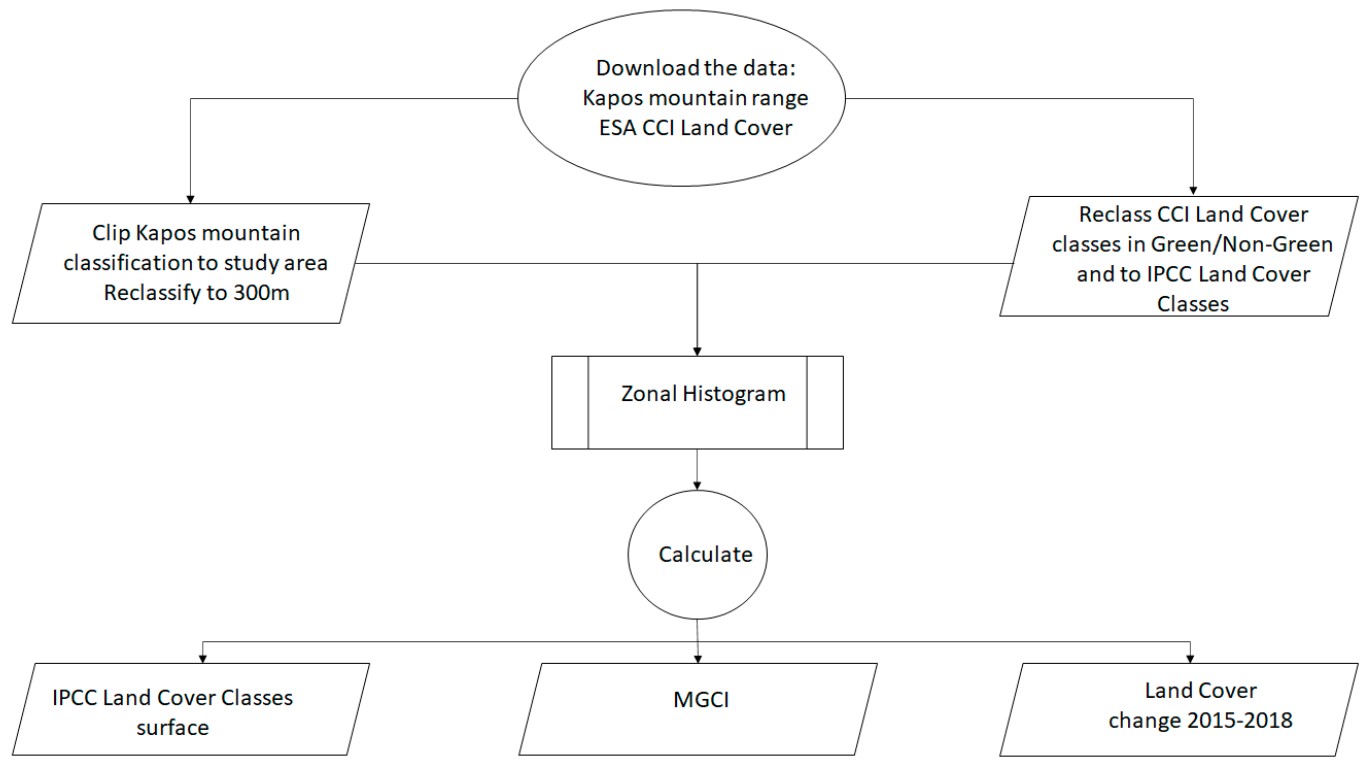

**Figure 1.** Workflow to calculate MGCI from EO products.

### 2.5. Innovation in the New FAO Method

The new FAO method has been compared to an existing one based on Collect Earth.

The first difference lies in methodology behind the process of land cover classification. Collect Earth relies on the visual interpretation of very high-resolution images provided by Google Earth Engine.

By contrast, the new methodology is based on the use of readily available land cover products which are the result of quantitative analysis of spectral and phenological features of medium-resolution satellite images, via a supervised approach based on global in situ reference data. While the former approach leaves space for subjective interpretation, the second is quantitative.

The second difference lies in the type of outputs. Collect Earth provided MGCI statistics in tabular format at national and at elevation zone levels. While this is sufficient to monitor the indicator over time at the two levels, it does not provide information about the local variability within the strata and the possibility to extract and identify hot spots. The new method delivers spatially explicit MGCI data (maps) from which tabular statistics are extracted at national and elevation zone levels. The spatially explicit output allows one to pinpoint where the mountain green cover areas are within a country and where the transitions are taking place. This can be extremely valuable for decision makers and the public.

Thirdly, from an operational point of view, the Collect Earth method is resource intensive, requiring a large team to set up the Collect Earth environment, to design the survey, to interpret and validate user inputs, and to finally harmonization the results.

The new method is resource conservative: it avoids altogether the intense process of land cover classification as it relies on existing LC standardized products. The new method simply adds a layer of reclassification and synthesis. The availability of the ESA and Copernicus LC products through the Google Earth Engine platform allows for easy deployment via open source software with minimum effort and virtually zero maintenance costs.

### 2.6. Accuracy Assessment

We measured the accuracy of the new MGCI data, by comparing the green/non-green maps obtained, respectively, from the ESA CCI-LC maps and from the CGLC-LC100 maps for 2015 against a global validation dataset and a national one. The global validation dataset was obtained from the work carried out by Li et al. in 2017 [14].

The global dataset [15] contains 36,000 samples with land cover ground-truth labels. The data were collected globally through the visual interpretation of images from the Landsat 8 data archive for 2015, conducted by a team of expert interpreters. The ground-truth dataset uses a land cover classification system built with 11 level-1 classes and 28 level-2 classes; the full legend is available in Table A2.

The national dataset contains 1580 ground-truth land cover samples, collected in the field by the FAO in Lesotho in May 2021, under the national land cover mapping framework. The ground-truth dataset uses a land cover classification system built using the FAO Land Cover Classification System, and the full legend is available in Table A3.

For the validation of the MGCI, each data point in the global and the national validation datasets was reclassified into either green or non-green, and only points falling within mountain zones were retained, resulting in a final dataset of 8109 validation samples for the global dataset and 1386 for Lesotho.

We thus calculated a confusion matrix by comparing the green/non-green cover map derived from the ESA CCI-LC and CGLS-LC100 maps, respectively, against the global and national ground-truth land cover data points relabeled either as green or non-green. The classification accuracy was assessed using the producer's accuracy (PA), the user's accuracy (UA), the overall accuracy (OA), and the kappa coefficient based on the confusion matrix. PA is the probability that a pixel is correctly classified as a land cover type and is the reciprocal of the error of omission. User's accuracy (UA), instead, indicates the proportion of pixels that are correctly classified within the image and represents the reciprocal of the errors of commission. The overall accuracy, then, is the ratio between the total number of correctly classified pixels versus the total number of pixels used for accuracy assessment.

The kappa coefficient estimates the coincidence of two variables, considering the degree of overlap that would be expected by chance alone. A kappa value of 1 represents perfect agreement, while a value of 0 represents no agreement. In this study, the kappa coefficient has been used to measure the agreement between classification and validation samples for green/non-green areas.

The kappa coefficient is calculated using the formula developed by Congalton in 1991 [16]:

$$K = \frac{\sum_{i=1}^{r} Xii - \sum_{i=1}^{r}(Xi + *X + 1)}{N2 - \sum_{i=1}^{r}(Xi + *X + 1)} \tag{2}$$

where $r$ is the number of rows in the matrix, $Xii$ is the number of observations in rows $i$ and column $i$, $Xi+$ and $Xi+1$ are the marginal totals of row $i$ and column $i$, respectively, and N is the total number of observations.

### 2.7. Global Accuracy Assessment of Green/Non-Green Cover Derived from ESA CCI-LC Map

The results of the accuracy assessment of green/non-green cover derived from ESA CCI against the global validation datasets are presented in Table 3. They yielded an OA of 86%, PA of 93.64%, and 59% for green and non-green classes, respectively. UA was 89% and 72.35%, respectively. Kappa was 0.56, which defines a good degree of agreement [17].

The green class was predicted with higher accuracy than the non-green class, the latter being mainly affected by omission errors, as indicated by the low PA. To understand the underlying reasons, 100 sites, associated with omission errors, were randomly selected. For each site, we compared very high-resolution images provided by Google and ESRI, the ESA CCI-LC map, and the GLC-LC100 map.

**Table 3.** Confusion matrix and accuracy assessment of green/non-green classification based on CCI-LC, 2015.

| | | Truth | | Total | User Accuracy | Kappa |
|---|---|---|---|---|---|---|
| | | Green | Non-Green | | | |
| **Predicted** | Green | 5924 | 731 | 6655 | 0.890 | 0 |
| | Non-Green | 402 | 1052 | 1454 | 0.723 | 0 |
| | Total | 6326 | 1783 | 8109 | 0 | 0 |
| | Producer's Accuracy | 0.936 | 0.590 | 0 | 0.860 | 0 |
| | Kappa | 0 | 0 | 0 | 0 | 0.563 |

From the very high-resolution images, it could be seen that the sites were characterized, in most cases, by bare land patches surrounded by sparse vegetation, as shown in column *a* in Figure A2. These areas were classified as shrubland, grassland, or sparse vegetation according to the CCI-LC dataset, as shown in column b in Figure A2.

The disagreement between ground-truth and the ESA CCI, stemming mainly from omission errors, can be explained by the compound effect of (i) errors in the CCI-LC product affecting the accuracy of sparse vegetation, shrubland, and grassland classes [8], and (ii) the low spatial resolution of the CCI-LC which does not allow it to discriminate landscape features that are small in size, such as bare land patches surrounded by larger green areas.

*2.8. Global Accuracy Assessment of Green/Non-Green Cover Derived from GLC-LC100*

The green/non-green cover derived from the CGLS-LC100 map scored an overall accuracy of 93.9%, with producer's accuracy of 99% and 74% and user's accuracy of 93.2% and 97.6%, respectively, for green and non-green classes. The kappa coefficient was 0.8, indicating excellent agreement between the two datasets (Table 4).

**Table 4.** Confusion matrix and accuracy assessment of green/non-green classification based on CGLS-LC100, 2015.

| | | Truth | | Total | User's Accuracy | Kappa |
|---|---|---|---|---|---|---|
| | | Green | Non-Green | | | |
| **Predicted** | Green | 6294 | 459 | 6753 | 0.932 | 0 |
| | Non-Green | 32 | 1324 | 1356 | 0.976 | 0 |
| | Total | 6326 | 1783 | 8109 | 0 | 0 |
| | Producer's Accuracy | 0.994 | 0.742 | 0 | 0.939 | 0 |
| | Kappa | 0 | 0 | 0 | 0 | 0.806 |

The use of GLC-LC100 resulted in considerable improvements in the accuracy of the green/non-green classification when compared to ESA CCI-LC. These improvements are likely due to: (i) the inherently higher accuracy of the GLC-LC100 product, at 80.6% [9], compared to the ESA CCI-LC (estimated at 75%); (ii) the GC-LC100 map's higher spatial resolution at 100 m, allowing discrimination of smaller patches of specific land cover types.

The superior ability of GLC-LC100 to discriminate small bare land patches was further confirmed by the comparison with the ESA CCI-LC at randomly selected test sites shown in column *c* in Figure A2. Sites in Argentina, the United States, Syria, and China, where satellite background images showed bare land patches surrounded by sparse vegetation, were classified as shrubland, cropland, or mosaic tree and shrubs according to the ESA CCI-LC. The GLC-LC100 was instead able to discriminate the patches of bare land or sparse vegetation from the surrounding green areas such as shrubs or herbaceous vegetation. Only in the case of the Syria test site did the ESA CCI-LC identify bare land in proximity to the ground-truth point.

In the test site in Chile, the background image showed snow covering mountains, and the ESA CCI-LC classified the area as "trees deciduous open", while the GLC-LC100 was able to discriminate a patch of bare land surrounded by herbaceous vegetation and by closed deciduous forest further away. In this case, it appears that the CCI-LC overestimated open forest, underestimated closed forest, and failed to discriminate bare land.

In the test site in Saudi Arabia, the background image showed a dominant pattern of bare land, and both the ESA CCI-LC and GLC-LC100 maps correctly classified the site as bare land. In the test site in Kenya, lastly, the CCI-LC classified a patch of bare land as grassland whereas the GLC-LC100 classified it as herbaceous vegetation, both indicating erroneous results, with a higher tendency of the ESA CCI-LC to overestimate green vegetation.

### 2.9. National Accuracy Assessment of Green/Non-Green Cover Derived from ESA CCI-LC

The green/non-green cover map derived from the ESA CCI baseline for 2019 was used for this test, to minimize the time difference with the validation dataset (2021). We clipped the green/non-green map to the mountain area of Lesotho. The national subset map scored an overall accuracy of 99%, with producer's accuracy of 99.3% and 95.0% and user's accuracy of 99.6% and 92.15%, respectively, for green and non-green classes. The kappa coefficient was 0.84, indicating excellent agreement between the two datasets (Table 5).

**Table 5.** Confusion matrix and accuracy assessment of green-non green classification based on ESA CCI 2019.

|  |  | Truth | | Total | User's Accuracy | Kappa |
|---|---|---|---|---|---|---|
|  |  | Green | Non-Green |  |  |  |
| **Predicted** | Green | 1175 | 4 | 1179 | 99.660 | 0 |
|  | Non-Green | 8 | 94 | 102 | 92.156 | 0 |
|  | Total | 1183 | 98 | 1281 | 0 | 0 |
|  | Producer's Accuracy | 99.323 | 95.918 |  | 0.990 | 0 |
|  | Kappa | 0 | 0 | 0 | 0 | 0.849 |

### 3. Results

The new MGCI calculation method introduced by the FAO in 2020 allowed us to compute, for the first time, the Mountain Green Cover Index over time as well as monitor land cover changes in mountain areas. Despite the higher accuracy of the GLC-LC100 map highlighted in the previous section, in 2020, the FAO primarily relied on ESA CCI-LC maps for two main reasons: firstly, as explained above, only the ESA CCI-LC maps provided a historical time series ranging from the year 2000 to 2018, as is customary for similar indicators (e.g., SDG indicators 15.1.1 and 15.2.1). Secondly, the full range of GLC-LC100 maps was not yet publicly available at the time in which the new MGCI figures needed to be calculated according to the established annual SDG reporting cycle. By adopting the ESA CCI-LC as the default data source, therefore, the MGCI figures updated in 2020 were affected by the same main accuracy limitations as their source product, which should be kept in mind when interpreting the results below.

### 3.1. Spatial/Temporal Distribution of MGCI

Time series data dating back to the year 2000 revealed a small overall percentage change to the global MGCI, with an initial period of expansion between 2000 and 2010, followed by a period of retraction between 2010 and 2018, as shown in Figure 2. While the decline in the global MGCI between 2010 and 2018 may appear small in terms of percentage change (approximately 0.04%), this represents a loss of mountain green cover equivalent to 7510.77 km$^2$

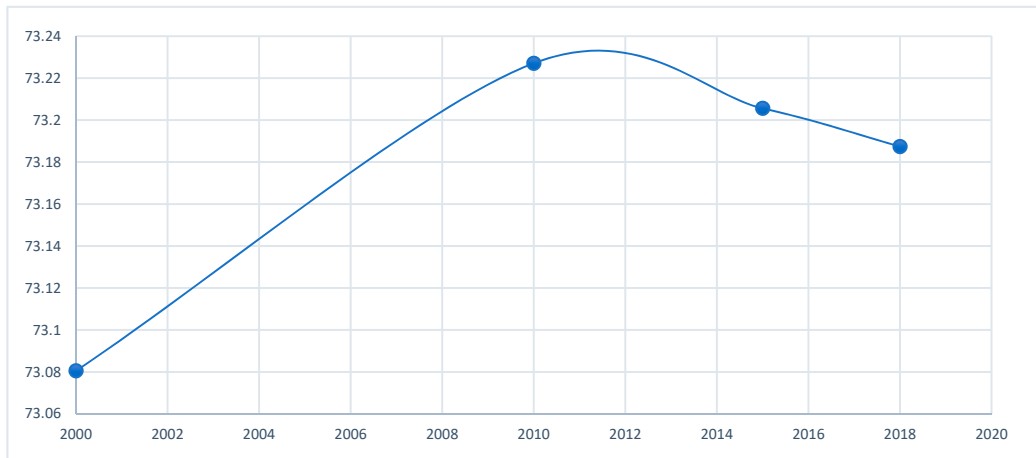

**Figure 2.** MGCI assessed from CCI-LC over time. Data points include 2000, 2010, 2015, and 2018. Trend has been smoothed using a spline function.

Figure 3 shows that in 2018, East and South East Asia had the highest proportion of green mountain cover, at 87%, while West Asia and North Africa had the lowest cover, at 63%. Oceania and Latin America and the Caribbean had a green mountain cover of 86 percent and 82%, respectively, followed by Sub-Saharan Africa at 80% and Australia and New Zealand at 78%. North America and Europe and Central and South Asia had green mountain cover between 69% and 68%.

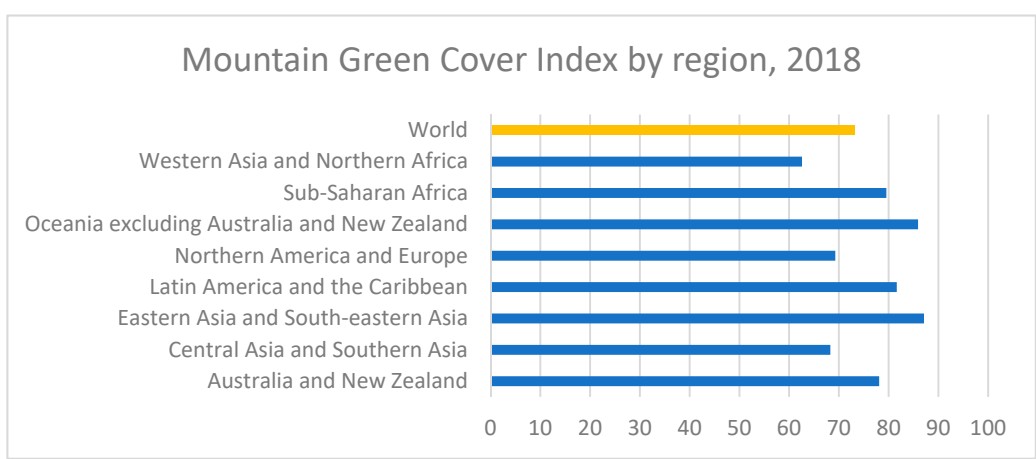

**Figure 3.** Mountain Green Cover Index by regions as defined by the System of Integrated Environmental and Economic Accounting (SEEA).

Disaggregated data by land cover type and elevation, shown in Figure 4, revealed important patterns for the world's mountains.

- Forest: At the lowest elevation, forests are the predominant land cover type, covering over 50 percent of the area. As expected, however, the share of forest cover steadily drops with higher elevation, becoming almost negligible above 4500 m.
- Grassland and other land: The proportion of mountain area covered by grassland and other land (which may include ice cover, glaciers, and barren land) generally increases with elevation, with grassland appearing to be the predominant land cover type above 3500 m.
- Cropland: Across elevation ranges, cropland is most expanded between 1500 and 2500 m, probably reflecting the fact that mountains at lower elevations are also defined by a higher slope and local elevation range (LER), which may not provide a suitable landscape for growing crops, whereas elevation ranges between 1500 and 2.500 offer

greater possibilities of harboring plateaus. Above 2500 m, crop coverage of mountains also steadily decreases.

- Settlement and wetland: The share of mountain cover of settlements and wetland is negligible at all elevation ranges, although also with a tendency to decrease with higher altitudes.

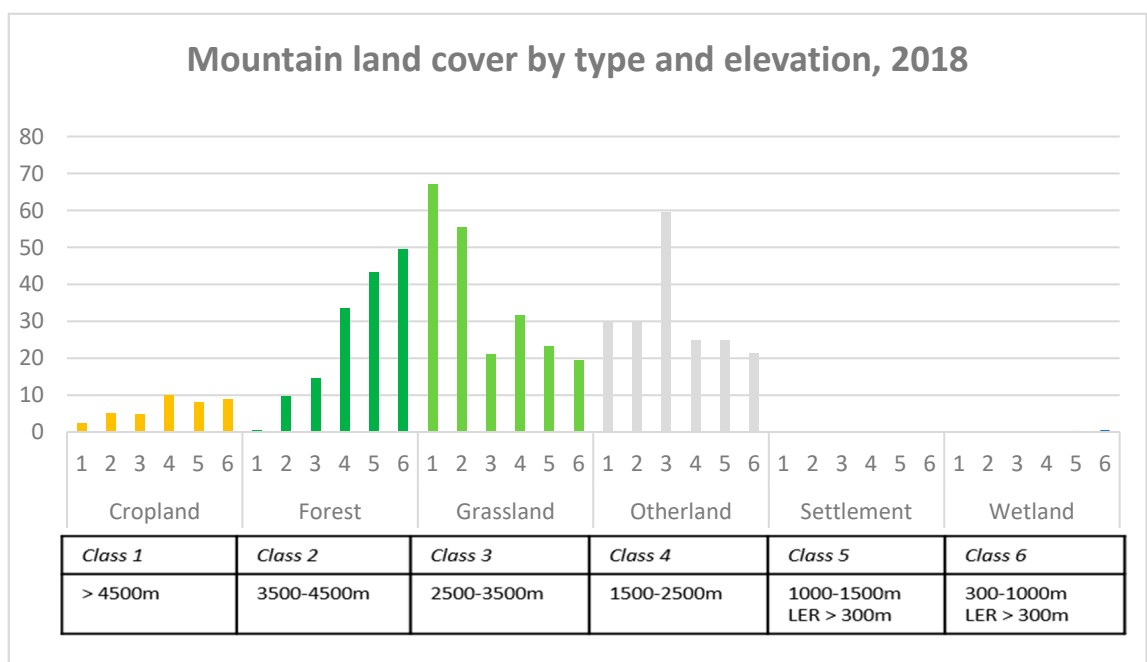

**Figure 4.** Distribution of land cover types by elevation class in 2018.

### 3.2. Comparison with FAO 2017 Baseline (Based on Collect Earth)

MGCI estimates were calculated at a national level from the ESA CCI-LC for the year 2017 and compared with MGCI estimates calculated by the FAO in 2017 using Collect Earth. The two datasets were analyzed using a *t*-test assuming equal variance as shown in Figure 5. Results showed that the difference between the means of the two distributions was not statistically significant (P = 0.51). The high consistency between the dataset was further confirmed by linear regression yielding an $R^2$ = 0.91. This test confirms the consistent quality of the CCI-LC estimates and indicates that both methodologies are valid for assessing the status of the indicator. The benefit of using the wall-to-wall approach (ESA CCI-LC) becomes evident when comparing the changes (which systematic sampling could not detect well as they were rare), time required for the data collection, and validation, which is simpler.

### 3.3. MGCI Trends and Land Cover Trends

To obtain better insights into the land use/land cover dynamics in mountainous areas, a trend analysis using the reclassified ESA CCI layers was carried out for the period 2015–2018. Countries were categorized in three groups depending on the MGCI trend, as shown in Figure 6.

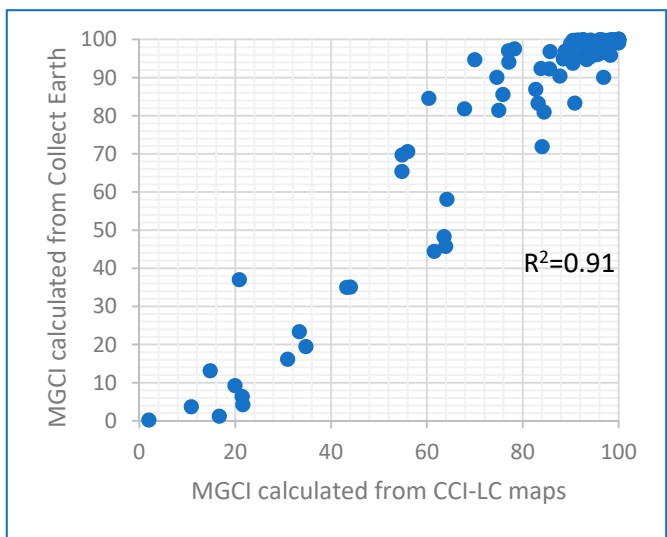

**Figure 5.** Scatterplot of MGCI estimates calculated for countries using CCI-LC and Collect Earth in 2017, showing that the two datasets are highly correlated.

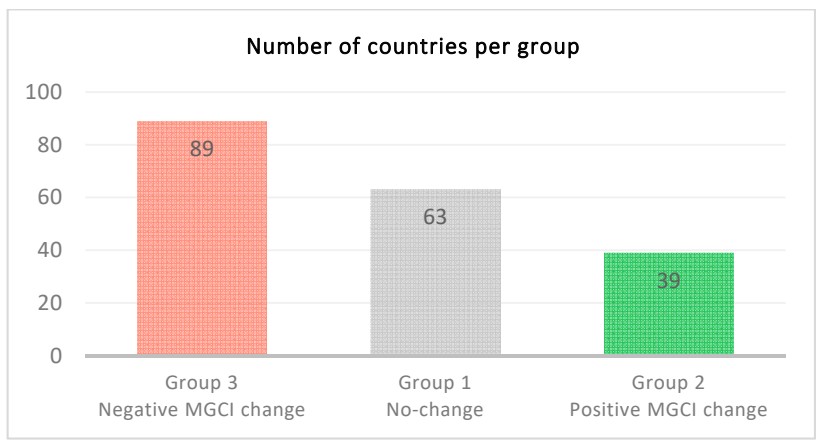

**Figure 6.** Number of countries experiencing negative, positive, or no change in MGCI from 2015 to 2018.

- Group one included countries (63) which experienced no significant absolute or relative changes in the MGCI during the period 2015–2018. Of these, forest cover increased mostly in Equatorial Guinea (+9 Km$^2$), Côte D'Ivoire (+12.33 Km$^2$), Togo (+12.44 Km$^2$), Nicaragua (+45 Km$^2$), and Republic of the Congo (+61 Km$^2$). Expansion of forest cover was mainly counterbalanced by contraction of cropland in Congo and Equatorial Guinea and mainly of grassland in Nicaragua, Togo, and Côte d'Ivoire, as shown in Figure 7.
- Group two included countries (39) that experienced a positive change in the MGCI during the period 2015–2018. On average, the MGCI increased by +0.11%, with the largest gain in Niger (+0.75%). The total mountain green area across the entire group increased by +3197 Km$^2$. Group level analysis of land cover change in mountain areas indicated a net loss of forest cover, and net gains of grassland, cropland, and wetland, as shown in Figure 8. Green cover encroached into other land areas, triggering the positive change in the MGCI.
- Between 2015 and 2018, the Russian Federation lost 10185 Km$^2$ of forest cover in mountain areas, accounting for a 96% net loss for group 2. It gained 5560 Km$^2$ and 4494 Km$^2$ of grassland and cropland, respectively. However, these figures should be interpreted with caution as forest cover is often affected by factors that do not result

in a permanent land cover/land use change. These factors include fires, other natural or anthropogenic disturbances, and forest management. In fact, it has been recently reported that forest fires explained 50–60% of tree cover loss in boreal forests from 2003–2018 [18]. The four largest forest cover gains occurred in Ethiopia (2772 $Km^2$), Mongolia (2347 $Km^2$), Peru (433 $Km^2$), and Kyrgyzstan (414 $Km^2$), together accounting for 80% of the total forest mountain cover net gain of the group. Figure 9 illustrates the land cover changes in the five countries.

- Group three included countries (89) where the MGCI decreased in the period 2015–2018. The MGCI decreased on average by 0.04%, with the largest loss in Tajikistan (−0.6%). The moderate changes indicate that total mountain green cover did not significantly decrease in these countries. Group-level land cover change analysis indicated a net loss of green vegetation (−361 $Km^2$). Significant changes in land cover classes in mountain areas were recorded at the group level, as shown in Figure 10. Forest cover was lost in 30 countries (−8513 $Km^2$) and was gained in 54 countries (+20,732 $Km^2$), resulting in a net gain of +12,219 $Km^2$. Grassland cover was lost in 57 countries (−19,929 $Km^2$) and gained in 28 countries (+5005 $Km^2$), for a net loss of −14,924 $Km^2$. Cropland was lost in 57 countries (−9703 $Km^2$) and gained in 31 countries (+2344 $Km^2$) for a net loss of 7370 $Km^2$. Bare cover increased in 47 countries (13,516 $Km^2$) and decreased in three countries (−223 $Km^2$). Settlements increased in almost all countries (+3666 $Km^2$). Figure 11 shows the three countries with highest forest cover gains and the three countries with highest forest cover losses. China, Turkey, and India gained 8050 $Km^2$, 1904 $Km^2$, and 1709 $Km^2$ of forest cover, respectively. In all three cases, a combined loss of grassland and cropland overcompensated the forest cover gain, producing an overall net loss of green cover. China experienced the highest increase in both bare land cover (of 5045 $Km^2$) and settlement cover (of 617 $Km^2$).

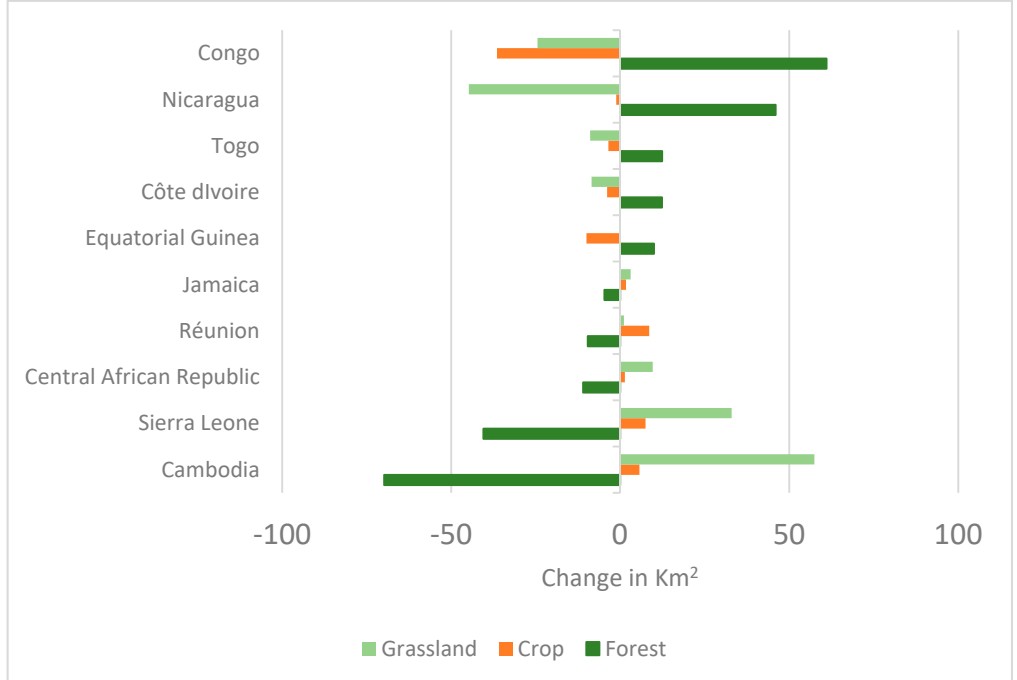

**Figure 7.** Land cover changes in mountain areas in selected countries where MGCI did not change from 2015 to 2018.

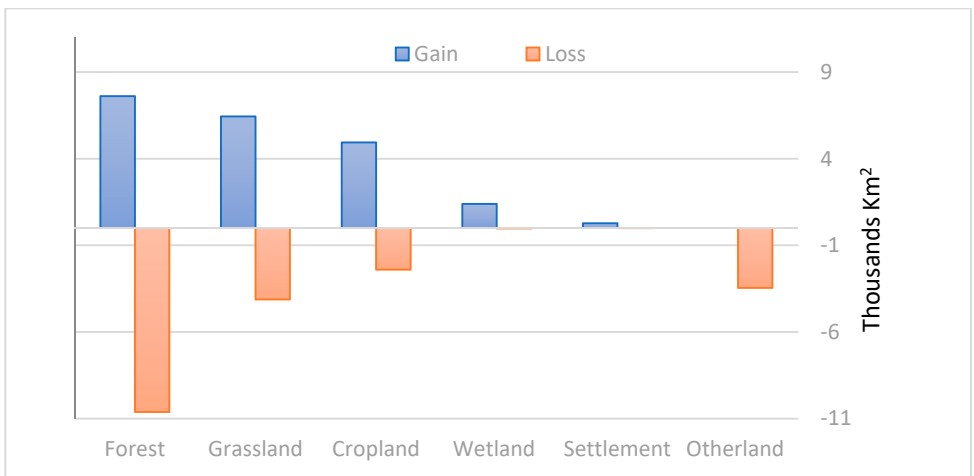

**Figure 8.** Total gains and losses of land cover in mountain area per class in countries where MGCI increased from 2015 to 2018.

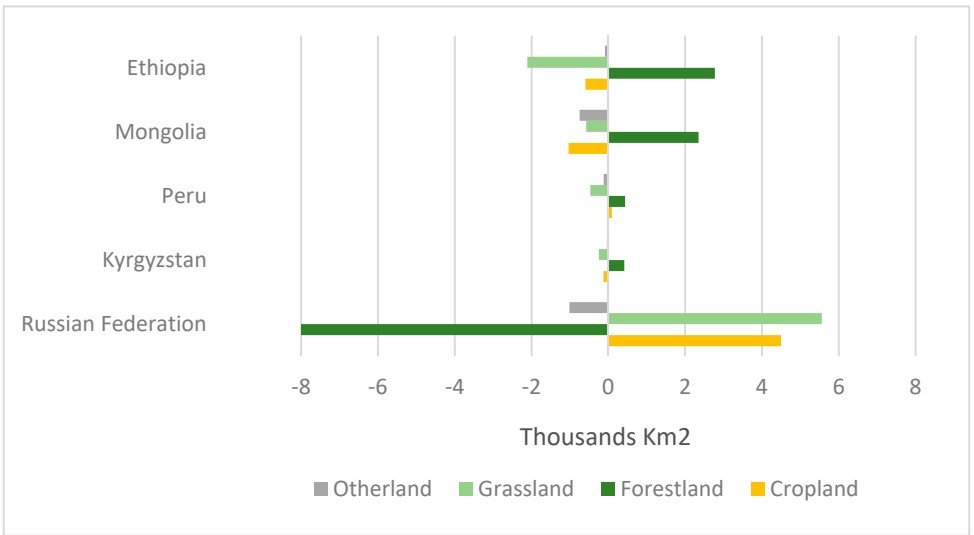

**Figure 9.** Land cover changes in mountain areas in selected countries experiencing increase in MGCI from 2015 to 2018.

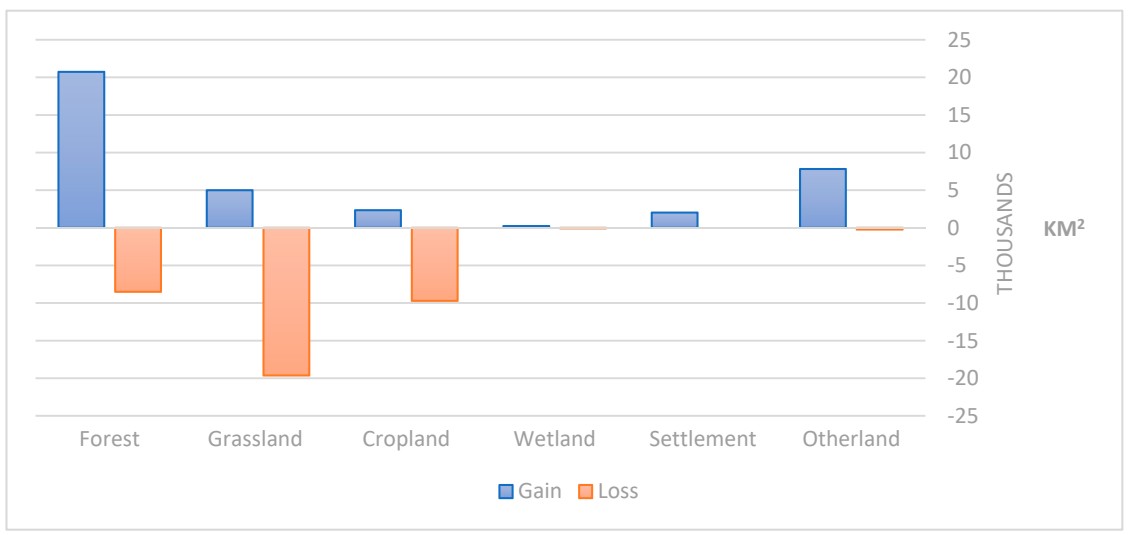

**Figure 10.** Gain and loss of land cover classes in mountain areas in countries where MGCI decreased from 2015 to 2018.

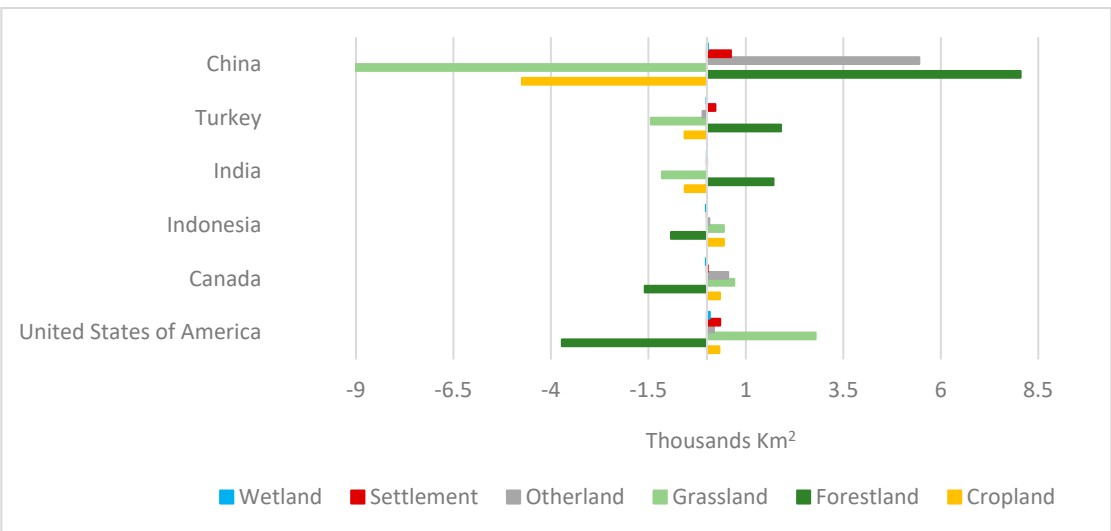

**Figure 11.** Land cover changes in mountain areas in selected countries where MGCI decreased from 2015 to 2018.

The United States, Canada, and Indonesia suffered a loss of forest cover ($-3720$ Km$^2$, $-1592$ Km$^2$, and $-919$ Km$^2$, respectively). In Indonesia, the loss of forest cover was compensated by an increase in cropland and grassland cover. In Canada, the forest cover loss was compensated by increases in grassland cover (44%), cropland cover (21%), and by expansion of other land (34%). In the United States, three quarters of the forest cover lost was taken by the expansion of grassland, whereas 9% was taken by expanding cropland. The residual area was taken by the expansion of settlements (9%), other land (5%), and wetland (2%) cover. Again, the changes from forest cover to other green cover classes should be interpreted with care as they do not necessarily reflect permanent cover changes.

### 3.4. Introducing Weights into the MGCI Formula

The combined land cover and MGCI analysis over the period 2015–2018 highlighted a limited sensitivity of the indicator to land cover change dynamics. For instance, in Congo, where forest cover increased, and in Cambodia, where forest cover decreased, the MGCI was stable over the study period (Figure 7). In Russia, where there was a forest cover loss, the MGCI increased (Figure 9). In Turkey and in India, there were forests gains, however, the MGCI decreased (Figure 11).

The incoherence between the MGCI trend direction and the forest cover change directions is rooted in the definition of the mountain green cover area in the numerator of the MGCI formula:

$$MGCI = \frac{Mountain\ Green\ Cover\ Area}{Total\ Mountain\ Area} \tag{3}$$

where *Mountain Green Cover Area* = Forestland$_{area}$ + Cropland$_{area}$ + Grassland$_{area}$ + Wetland$_{area}$.

The four vegetation classes have equal importance, therefore, a given area gain or a loss in any of the four classes would produce the same impact on the final MGCI outcome. Furthermore, an area gain or a loss in one class that is compensated by an equal loss or gain, respectively, in another class, would result in no change in the total mountain green cover area and hence would result in no impact on the final MGCI outcome.

Taking an extreme case, if in a country the entire forest cover in mountain areas was converted to cropland, the MGCI indicator, measured before and after the land cover change, would be the same. It is patent that the MGCI, in its current definition, is not able to recognize the negative impact on mountain ecosystem health caused by the encroachment of forests due to agricultural expansion. More specifically, the MGCI indicator falls short in recognizing the individual contribution that different vegetation classes provide to mountain ecosystem health.

One way to address this issue could consist in weighting the land categories under consideration based on relative value towards achieving SDG 15.4, as shown in Equation (2) below:

$$WMGCI = \frac{Forestland_{area} * W_f + Cropland_{area} * W_c + Grassland_{area} * W_g + Wetland_{area} * W_w}{Total\ Mountain\ Area} \quad (4)$$

where $W_f$, $W_c$, $W_g$, $W_w$, are weights assigned, respectively, to forest, cropland, grassland, and wetland.

The weights ranged from 0.5 to 1, with 1 for the vegetation class with maximum relevance to ecosystem health and 0.5 for the vegetation class with the lowest relevance. The weights were empirically inferred, as shown in Table 6, on the assumption that (i) forest and wetland have the highest relevance, (ii) grassland has a lower relevance than forest but higher relevance than cropland.

**Table 6.** Vegetation-specific weights used to calculate the WMCGI (forest = 1, cropland = 0.5, grassland = 0.8, wetland = 1).

| Weight | Value |
|---|---|
| $W_f$ | 1 |
| $W_c$ | 0.5 |
| $W_w$ | 0.8 |
| $W_w$ | 1 |

WMGCI was calculated for four countries (Cambodia, Russia, China, and USA) for the years 2015 and 2018 and results were compared with MGCI estimates for the same period, as shown in Table 7.

**Table 7.** MGCI and WMGCI calculated for 2015 and 2018 and trend analysis for Cambodia, Russia, USA, and China.

| | MGCI | | | | WMGCI | | | |
|---|---|---|---|---|---|---|---|---|
| **Country** | **2015** | **2018** | **Delta** | **Trend** | **2015** | **2018** | **Delta** | **Trend** |
| Cambodia | 99.99423 | 99.99423 | 0.00 | Stable | 99.99423 | 97.29706 | −2.69717 | Down |
| Russia | 83.07241 | 83.10047 | 0.03 | Up | 0.783961 | 0.783299 | −0.00066 | Down |
| USA | 84.07327 | 84.03938 | −0.03 | Down | 84.07327 | 74.87713 | −9.19614 | Down |
| China | 84.80503 | 84.5641 | −0.24 | Down | 84.80503 | 69.44609 | −15.3589 | Down |

Overall, WMGCI estimates were lower than MGCI ones for all countries. This is explained by weights lower than 1 assigned to cropland and rangeland. The WMGCI change between 2015 and 2018 in Cambodia and in Russia was negative as opposed to the MGCI change that was, respectively, null and positive. The WMGCI change in the USA and in China was very significantly negative (−9% and −15%, respectively) while the MGCI change was just below 0.

In conclusion, the introduction of weights introduced sensitivity into the MGCI indicator to changes in the internal composition of the vegetation cover, and enhanced its capacity to account for the higher ecosystem value of natural vegetation in comparison to agriculture.

## 4. Discussion

This paper presents a first evaluation of an earth observation-based methodology to derive regular information on SDG indicator 15.4.2, the Mountain Green Cover Index (MGCI), solely from open access land cover data using an automated algorithm that avoids the need for visual interpretation. As such, this new approach provides countries with a methodology that they can easily use to validate the national estimates generated by the FAO, or alternatively, a template that countries can easily reuse to generate their own MGCI

values, for instance, by replacing the global default land cover maps with national maps. The study thoroughly assesses the accuracy of the EO methodology, and describes the main results at global, regional and national levels that capture critical land cover dynamics that could be beneficial or detrimental to the health of mountain ecosystems.

The new default calculation method introduced by the FAO in 2020 proved to be robust when assessing the status of the mountain land cover, with an overall accuracy of 86% when using ESA CCI-LC data at a 300 m resolution, and 93% when using the newer GLC-LC100 data at a 100 m resolution. The inherent accuracies of these land cover products were reflected in the accuracy of the MGCI estimates generated therefrom. However, it has to be noted that even in cases where two independent land cover classifications are highly accurate, the accuracy of a change map (or estimate) produced using them can be low and the result biased [19].

Relying on global datasets maintained by the ESA CCI and the Copernicus Land service ensures long-term sustainability of the proposed solution, and the possibility to automatize the workflow using open source EO platforms such as Google Earth Engine or open and free GIS software such as QGIS. Furthermore, the new approach easily accommodates future improvements in global land cover products that would further improve the accuracy of MGCI estimates. In this regard, it is noted that the World Cover Project sponsored by the European Space Agency is currently prototyping a global 10 m resolution global land cover map that is expected to be updated every three months (https://esa-worldcover.org/en, accessed on 15 December 2020).

Compared to the previous default data collection method for the MGCI, which was based on visual interpretation of a selected number of sample sites across the globe, the new approach is less resource intensive and more straightforward, and therefore increases the likelihood that National Statistical Offices or specialized national geospatial units will use it. The main results from the MGCI validation process led by the FAO in 2020 confirm the positive impact of the introduction of the methodology: 39% of countries contacted responded to the validation request; 21% of countries approved the validation request, as opposed to 12% in 2017; and 13% provided their own national data, as opposed to zero such cases in 2017.

Lastly, the study extensively assessed the sensitivity of the MGCI to land cover change dynamics that could be beneficial or detrimental to mountain ecosystem health, for the period 2015–2018. Results showed that the aggregate MGCI was not sensitive to cases of forest encroachment due to agriculture expansion in the Central African Republic, Sierra Leone, or Cambodia. This highlighted the importance of disaggregating the indicator by land cover type to monitor trends across the various green and non-green land cover classes. In Russia, the aggregate MGCI improved over the reference period, yet disaggregated data suggested land cover changes within the mountain green cover. However, such land cover dynamics detected by the ESA CCI data may in fact be the result of forest fires and forest management, hence further investigation should be carried out. In China, Turkey, and India, the aggregate MGCI decreased despite the gains in forest cover which were overshadowed by larger losses in cropland and grassland. To improve the sensitivity of the MGCI indicator, its original formula was modified by introducing vegetation-specific weights in the numerator, resulting in the Weighted MGCI (WMGCI). The WMGCI was tested in five countries. Results showed that the WMGCI (i) was sensitive to changes in the composition of the vegetation cover, and (ii) could better account for losses or gains of natural vegetation. In conclusion, the WMGCI was more closely related to the status of natural vegetation and hence to the ecosystem health of mountain ecosystems.

The approach proposed by the FAO allows the measurement and monitoring of the MGCI through the analysis of land cover maps jointly with the mountain elevation layer. The work is based on a straightforward GIS routine that can be easily implemented by countries using existing free and open software such as QGIS and platforms such as Google Earth Engine. The simplified methodology alleviates the burden on countries, while ensuring consistency and accuracy.

Limitations of the definition of the MGCI have been identified, substantiating the need for a modified and enhanced version to properly account for (i) green vegetation appearing in areas formerly occupied by glaciers and perennial snow as a result of global warming, and (ii) agriculture over steep slopes which may lead to soil erosion. In this regard, the FAO will work in 2021 on the enhancement of the indicator definition and validate the methodology through pilot projects in countries.

**Author Contributions:** Conceptualization, Lorenzo De Simone, Pietro Gennari, Dorian Navarro, Anssi Pekkarinen, and Javier de Lamo; methodology, Lorenzo De Simone and Anssi Pekkarinen; software, Lorenzo De Simone; validation, Lorenzo De Simone, Dorian Navarro, and Anssi Pekkarinen; formal analysis, Lorenzo De Simone, Dorian Navarro, and Javier de Lamo; investigation, Lorenzo De Simone; resources, Pietro Gennari; data curation, Lorenzo De Simone and Dorian Navarro; writing—original draft preparation, Lorenzo De Simone; writing—review and editing, Lorenzo De Simone, Dorian Navarro, Pietro Gennari, Anssi Pekkarinen, and Javier De Lamo; visualization, Lorenzo De Simone, Dorian Navarro, and Javier de Lamo; supervision, Pietro Gennari and Anssi Pekkarinen; project administration, Lorenzo De Simone; funding acquisition, Pietro Gennari. All authors have read and agreed to the published version of the manuscript.

**Funding:** This research was funded under the FMM sub-program "Improving country data for monitoring SDG achievements and informing policy decisions".

**Data Availability Statement:** Publicly available datasets were analyzed in this study. The data can be found here:ESA CCI-LC: [http://maps.elie.ucl.ac.be/CCI/viewer/download.php], accessed on 15 December 2020.GLC-LC100: [https://land.copernicus.eu/global/products/lc], accessed on 15 December 2020.Global Mountain Classification: [http://www.mp-discussion.org/gis/mountain_area/globsrmtkpos1-6R5mol.tif], accessed on 15 December 2020.Global Land Cover Validation Sample (v.1): [http://data.ess.tsinghua.edu.cn/data/temp/GlobalLandCoverValidationSampleSet_v1.xlsx], accessed on 15 December 2020.

**Acknowledgments:** We would like to acknowledge the technical support provided by Yuka Makino and Chiara Patriarca, Forestry Division, FAO.

**Conflicts of Interest:** The authors declare no conflict of interest.

**Appendix A**

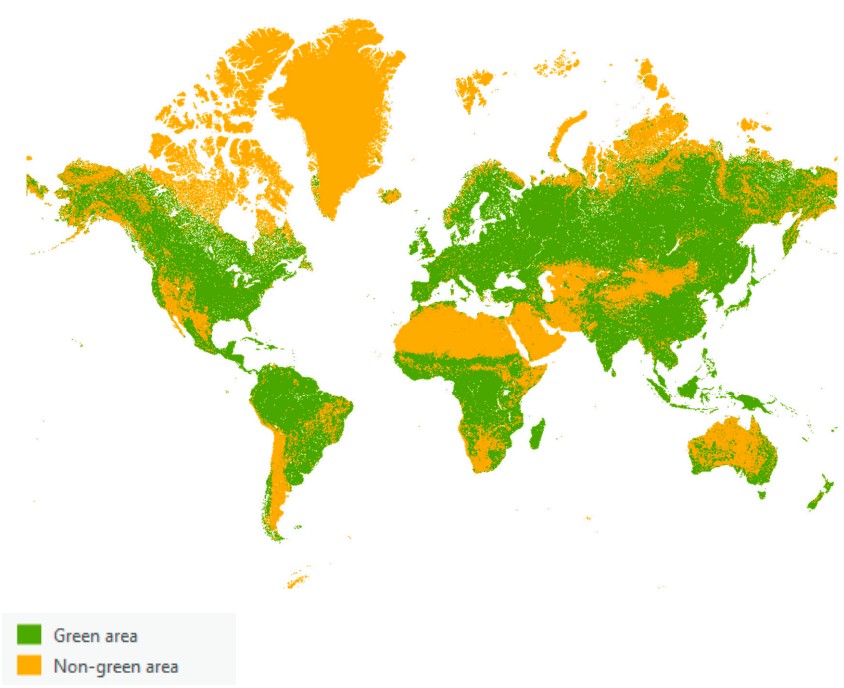

**Figure A1.** Green/non-green cover map obtained from reclassification of ESA CCI-LC 2015.

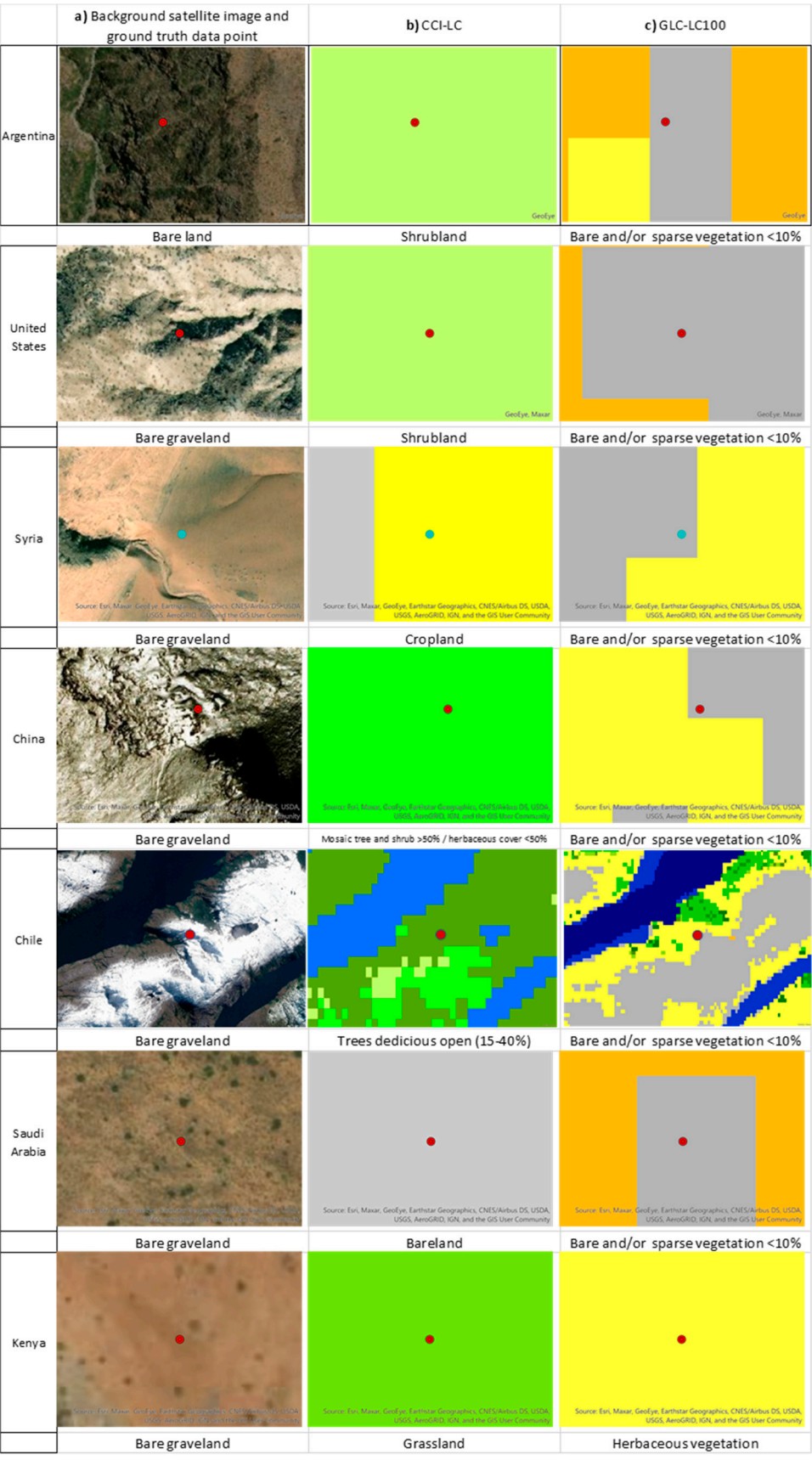

**Figure A2.** Selected sites associated with omission errors from the accuracy assessment of the green/non-green map: (**a**) background very high-resolution images from Google/ESRI; (**b**) land cover classified according to CCI-LC 2015; (**c**) land cover classified according to CGLS-LC100 2015.

**Table A1.** Land cover classes included in CGLS-LC100.

| Map Code | Land Cover Class | Definition According UN LCCS |
|---|---|---|
| 111 | Closed forest, evergreen needle leaf | Tree canopy > 70%, almost all needle leaf trees remain green all year. Canopy is never without green foliage. |
| 113 | Closed forest, evergreen deciduous needle leaf | Tree canopy > 70%, consists ofseasonal needle leaf tree communities with an annual cycle of leaf-on and leaf-off periods. |
| 112 | Closed forest, evergreen, broad leaf | Tree canopy > 70%, almost all broadleaf trees remain green year round. Canopy is never without green foliage. |
| 114 | Closed forest, deciduous broad leaf | Tree canopy > 70%, consists of seasonal broadleaf tree communities with an annual cycle of leaf-on and leaf-off periods. |
| 115 | Closed forest, mixed | Closed forest, mix of types. |
| 116 | Closed forest, unknown | Closed forest, not matching any of the other definitions. |
| 121 | Open forest, evergreen needle leaf | Top layer—trees 15–70% and second layer—mix of shrubs and grassland, almost all needle leaf trees remain green all year. Canopy is never without green foliage. |
| 123 | Open forest, deciduous needle leaf | Top layer—trees 15–70% and second layer—mix of shrubs and grassland, consists of seasonal needle leaf tree communities with an annual cycle of leaf-on and leaf-off periods. |
| 122 | Open forest, evergreen broad leaf | Top layer—trees 15–70% and second layer—mix of shrubs and grassland, almost all broadleaf trees remain green year round. Canopy is never without green foliage. |
| 124 | Open forest, deciduous broad leaf | Top layer—trees 15–70% and second layer—mix of shrubs and grassland, consists of seasonal broadleaf tree communities with an annual cycle of leaf-on and leaf-off periods. |
| 125 | Open forest, mixed | Open forest, mix of types. |
| 126 | Open forest, unknown | Open forest, not matching any of the other definitions. |
| 20 | Shrubs | These are woody perennial plants with persistent and woody stems and without any defined main stem, being less than 5 m tall. The shrub foliage can be either evergreen or deciduous. |
| 30 | Herbaceous vegetation | Plants without persistent stem or shoots above ground and lacking definite firm structure. Tree and shrub cover is less than 10%. |
| 90 | Herbaceous wetland | Lands with a permanent mixture of water and herbaceous or woody vegetation. The vegetation can be present in either salt, brackish, or fresh water. |
| 100 | Moss and lichen | Moss and lichen. |
| 60 | Bare/sparse vegetation | Lands with exposed soil, sand, or rocks and never have more than 10% vegetation cover during any time of the year. |
| 40 | Cultivated and managed vegetation/agriculture (cropland | Lands covered with temporary crops followed by harvest and a bare soil period (e.g., single and multiple cropping systems). Note that perennial woody crops will be classified as the appropriate forest or shrub land cover type. |
| 50 | Urban/built up | Land covered by buildings and other human-made structures. |
| 70 | Snow and ice | Lands under snow or ice cover throughout the year. |
| 80 | Permanent water bodies | Lakes, reservoirs, and rivers. Can be either fresh- or saltwater bodies. |
| 200 | Open sea | Oceans, seas. Can be either fresh- or saltwater bodies. |

**Table A2.** Land cover classes included in training dataset developed by Li et al.

| Level-1 | Level-2 | Description | Reclassification to Green/Non-Green |
|---|---|---|---|
| 10 Farmland | 11 Rice paddy<br>12 Greenhouse<br>13 Other farmland<br>14 Orchard<br>15 Bare farmland | | Green |
| 20 Forest | 21 Broadleaf, leaf-on<br>22 Broadleaf, leaf-off<br>23 Needle-leaf, leaf-on<br>24 Needle-leaf, leaf-off<br>25 Mixed leaf type, leaf-on<br>26 Mixed leaf type, leaf-off | Tree cover $\geq$ 10%;<br>Height > 5 m;<br>For mixed leaf, neither coniferous nor broadleaf types exceed 60% | Green |
| 30 Grassland | 31 Pasture, leaf-on<br>32 Natural grassland, leaf-on<br>33 Grassland, leaf-off | Canopy cover $\geq$ 20% | Green |
| 40 Shrublands | 41 Shrub cover, leaf-on<br>42 Shrub cover, leaf-off | Canopy cover $\geq$ 20%;<br>Height < 5 m | Green |
| 50 Wetland | 51 Marshland, leaf-on<br>52 Mudflat<br>53 Marshland, leaf-off | | Green |
| 60 Water | 61 Lake<br>62 Reservoir/ponds<br>63 River<br>64 Ocean | | Non-Green |
| 70 Tundra | 71 Shrub and brush tundra<br>72 Herbaceous tundra | | Green |
| 80 Impervious | 80 Impervious | | Non-Green |
| 90 Barren land | 90 Barren land | Vegetation cover < 10% | Non-Green |
| 100 Snow/Ice | 101 Snow<br>102 Ice | | Non-Green |
| 120 Cloud | 120 Cloud | | N/A |

**Table A3.** Land cover classes included in training dataset collected by FAO and Ministry of Agriculture and Food Security in Lesotho, 2021.

| Level-1 | Level-2 | Description | Reclassification to Green/Non-Green |
|---|---|---|---|
| Settlement | Urban<br>Rural | | Non-Green |
| Agriculture | Irrigated cropland<br>Rainfed cropland | | Green |
| Forest | Broadleaf<br>Needle leaf | Tree cover $\geq$ 10%;<br>Height > 5 m | Green |
| Herbaceous cover | Shrubland<br>Grassland<br>Wetland<br>Degraded grassland | Canopy cover $\geq$ 20%;<br>Canopy cover $\geq$ 20%;<br>Canopy cover $\geq$ 20%;<br>Canopy cover < 20% | Green<br>Green<br>Green<br>Non-Green |
| Bare surface | Gullies<br>Bare soil<br>Bare rock<br>Mining | | Non-Green |
| Water | River<br>Lakes | | Non-Green |

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
