# Peer review of "Using Standardized Time Series Land Cover Maps to Monitor the SDG Indicator “Mountain Green Cover Index” and Assess Its Sensitivity to Vegetation Dynamics"

_ijgi, doi:10.3390/ijgi10070427_

Round 1

Reviewer 1 Report

Dear Authors,

The manuscript is well documented, presenting a new aproach for calulation of one of the SDG Goals. 

Please go throughout the manuscript and correct spelling issues, misplaced comma or dots, misspelled units, unfinished sentences. Several examples are given below:

Line 18 “... that was superimposed on satellite images using.”

Line 21 “the limited involvement of counties 21 in estimating the MGCI”

459- 460 The total mountain green area across the entire 459 group increased by +3,197 Km2 Group level analysis of land cover change in

Line 491-509 Km 2 is written as km2 or Km2. Please correct althrough the manuscript

Line 636 “version. to properly account for”

Figure 2. MGCI assessed ....

Fig. 3 Mountain Green Cover Index ....

It is either Figure x. , either Fig. x

Best regards

Author Response

The following corrections were made:

Line 18 “... that was superimposed on satellite images using.” was changed to “... that was superimposed on satellite images.”

Line 21 “the limited involvement of counties 21 in estimating the MGCI” was changed to “the limited involvement of countries 21 in estimating the MGCI”

459- 460 "The total mountain green area across the entire 459 group increased by +3,197 Km2 Group level analysis of land cover change in" was changed to "The total mountain green area across the entire 459 group increased by +3,197 Km2. Group level analysis of land cover change in.."

Line 491-509 Km 2 is written as km2 or Km2. Please correct althrough the manuscript. Throughout the manscript Km was finally used. 

Line 636 “version. to properly account for” was changed to  “version to properly account for”

Figure 2. MGCI assessed ....

Fig. 3 Mountain Green Cover Index ....

It is either Figure x. , either Fig. x

Throughout the manuscript Figure x. was used.

Reviewer 2 Report

This is a very interesting paper that describes new landscape analysis method. It is well written and shows the method and the results well. From technical point of view I have only two comments bellow

  • Introduction is too long, in its current shape it looks like a short introduction with contributions section.
  • L36 why is this image here? And where is the caption

From more scientific points of view the authors failed to provides the reader with enough data on two pints, the level of innovation and the quality of results in comparison with current methods.

  1. In order to show the level of innovation it would make sense to have part where you compare existing methods from the technical point of view – part of is already in the text but it gets lost while reading so a condense paragraph would make sense
  2. In order to prove the method you need to compare the results with either real life data (from 1-2 selected areas) or other modeled data (from the same time). You partly did that but this is not enough to prove the quality of your work

Author Response

  • Introduction is too long, in its current shape it looks like a short introduction with contributions section.

We feel that the introduction needs to give enough emphasis to the SDG process, and to the previous work of FAO in the area of MGCI and the feedback from countries, which have led to the need for developing a new methodology. However introduction has been shortened as suggested.

  • L36 why is this image here? And where is the caption

This image was supposed to serve as a graphical abstract. After review we preferred to remove it.

From more scientific points of view the authors failed to provides the reader with enough data on two points, the level of innovation and the quality of results in comparison with current methods.

  1. In order to show the level of innovation it would make sense to have part where you compare existing methods from the technical point of view – part of is already in the text but it gets lost while reading so a condense paragraph would make sense

A short paragraph 2.4  has been added to explain briefly main difference between the new method and the Collect Earth one.

Differences in the land cover method, in the output and their application, and in the operations are provided.

  1. In order to prove the method you need to compare the results with either real life data (from 1-2 selected areas) or other modeled data (from the same time). You partly did that but this is not enough to prove the quality of your work

In order to meet this criteria, we have introduced a second validation level, at national level, using field data collected by FAO in 2021 in Lesotho during a national land cover survey. Such data was used to validate the Green-non Green cover derived from the ESA CCI 2019 baseline, which is the most recent data set available in the ESA time series.

Reviewer 3 Report

Review for the paper tiled: Using standardized time series land cover maps to monitor the 2 SDG indicator “Mountain Green Cover Index” and assess its 3 sensitivity to vegetation dynamics

Summary of Paper

The SDG indicators are pivotal in identifying how close or far each country is to achieving the wildly ambitious global SDG. Identifying best practices for uniformly collecting indicator data is a significant challenge. One that the authors of this paper are well suited to solve.

These authors are responding to the critique from global community and trying to improve existing methods for collecting one specific SDG indicator based critiques of the existing methods.

Value/Contribution of this paper:

The perspectives offered in this paper are unique and valuable since they are coming directly from the SDG indicator custodian agency. Their methods and justification are very strong, it is so important to document their decisions in the literature as they are doing here.

I think the most valuable point of this paper and EO applied to SDG, so papers like this one, is that it is addressing this issue mentioned on p 3

“However, not all countries have their own land cover map time series, and different land cover classification systems may exist at national level, which may require further adjustment of the data to comply with the standard classification used by the MGCI, i

Variables and datasets are well defined and well justified.

This paper offers value in the methodology but also in the results.

Specific comments and areas of improvement.

The first graphic is out of place and requires a figure caption.

What is not mentioned and could be valuable – the SDG indicators are aggregated at the country level. This method  - while these results are not shared here – could pin point where the Mountain Green Cover areas are within a country and where the transitions are taking place. This would be extremely valuable for decision makers and the public alike.

Style recommendations:

This is a well written article. It was a pleasure to read.  I do have a few minor recommendations.

Already in the abstract, very briefly tell the reader why SDGs are important and how EO data could help make it easier to monitor and evaluate progress on them. Later mention how EO data are freely available.

What is the MGCI and why is it important? What does it measure?

Formatting:

Title does not need a period

P 1 line 15 – too many spaces between : and SDG – line 18 too – maybe it is the justified formatting. Will stop pointing this out.

P3 line 76 open ( and never closed. Delete it.

What does MGCI stand for? Don’t see it spelled out the first time you use it other than in the title and abstract – please spell it again in main paper.

First figure has no caption, no number and is out of place. Question about the figure - Why use QGIS and ArcGIS?

Aren’t SDG indicator data updated over time? So the figure is confusing with the overtime part – is time in the calculation? Again, the figure is out of place – move it to a place where you describe it.

Line 164 page 4 – Kapos citation is referenced as 11 not as 3 as in other places – did the same author – author both publications?

Line 3345 – extra space? (several places - look carefully) 

Author Response

Specific comments and areas of improvement.

The first graphic is out of place and requires a figure caption.

Image was supposed to be a graphical abstract but n the end we preferred to remove it 

What is not mentioned and could be valuable – the SDG indicators are aggregated at the country level. This method  - while these results are not shared here – could pin point where the Mountain Green Cover areas are within a country and where the transitions are taking place. This would be extremely valuable for decision makers and the public alike.

Point has been addressed in a newly added paragraph 2.4 in which differences between new and old methods are compared. In this context the fact that the new methods allows for disaggregating the indicator at elevation zone level and below, and the spatially explicit nature of the output are explained.

Style recommendations:

This is a well written article. It was a pleasure to read.  I do have a few minor recommendations.

Already in the abstract, very briefly tell the reader why SDGs are important and how EO data could help make it easier to monitor and evaluate progress on them. Later mention how EO data are freely available.

Reference to the importance of SDG has been added at the beginning of the abstract. Advantage from use of EO data is explained.

What is the MGCI and why is it important? What does it measure? MGCI importance has been highlighted in the abstract as well as what it measures from a conceptual point of view. The quantitative information of what it measures are provided in paragraph 2.1

Formatting:

Title does not need a period

OK

P 1 line 15 – too many spaces between : and SDG – line 18 too – maybe it is the justified formatting. Will stop pointing this out.

ok

P3 line 76 open ( and never closed. Delete it.

ok

What does MGCI stand for? Don’t see it spelled out the first time you use it other than in the title and abstract – please spell it again in main paper.

MGCI  has been spelled out in the Introduction and in the Material and Methods additionally.

First figure has no caption, no number and is out of place. Question about the figure - Why use QGIS and ArcGIS?

 First figure was supposed to be a graphical abstract, but we preferred to delete it all together

Aren’t SDG indicator data updated over time? So the figure is confusing with the overtime part – is time in the calculation? Again, the figure is out of place – move it to a place where you describe it.

Line 164 page 4 – Kapos citation is referenced as 11 not as 3 as in other places – did the same author – author both publications?

Citation 3 refers to the authors who developed the definition of the mountain range (Kapos is the name of the main contributor). Citation 11 instead refers to the source where the Kapos dataset can be downloaded from. Reference has been corrected in page 4 L218

Line 3345 – extra space? (several places - look carefully) 

Removed all double spaces